# Complete Chloroplast Genomes of *Fagus sylvatica* L. Reveal Sequence Conservation in the Inverted Repeat and the Presence of Allelic Variation in NUPTs

**DOI:** 10.3390/genes12091357

**Published:** 2021-08-29

**Authors:** Bartosz Ulaszewski, Joanna Meger, Bagdevi Mishra, Marco Thines, Jarosław Burczyk

**Affiliations:** 1Department of Genetics, Faculty of Biological Sciences, Kazimierz Wielki University, Chodkiewicza 30, 85-064 Bydgoszcz, Poland; warmbier@ukw.edu.pl (J.M.); burczyk@ukw.edu.pl (J.B.); 2Senckenberg Biodiversity and Climate Research Centre, Senckenberganlage 25, 60325 Frankfurt am Main, Germany; bagdevi.mishra@senckenberg.de (B.M.); marco.thines@senckenberg.de (M.T.); 3Department of Biological Sciences, Institute of Ecology, Evolution and Diversity, Goethe University, Max-von-Laue-Str. 13, 60483 Frankfurt am Main, Germany; 4LOEWE Centre for Translational Biodiversity Genomics, Georg-Voigt-Str. 14-16, 60325 Frankfurt am Main, Germany

**Keywords:** complete chloroplast genome, microsatellite, SNP, indel, heteroplasmy, population genomics, European beech

## Abstract

Growing amounts of genomic data and more efficient assembly tools advance organelle genomics at an unprecedented scale. Genomic resources are increasingly used for phylogenetic analyses of many plant species, but are less frequently used to investigate within-species variability and phylogeography. In this study, we investigated genetic diversity of *Fagus sylvatica,* an important broadleaved tree species of European forests, based on complete chloroplast genomes of 18 individuals sampled widely across the species distribution. Our results confirm the hypothesis of a low cpDNA diversity in European beech. The chloroplast genome size was remarkably stable (158,428 ± 37 bp). The polymorphic markers, 12 microsatellites (SSR), four SNPs and one indel, were found only in the single copy regions, while inverted repeat regions were monomorphic both in terms of length and sequence, suggesting highly efficient suppression of mutation. The within-individual analysis of polymorphisms showed >9k of markers which were proportionally present in gene and non-gene areas. However, an investigation of the frequency of alternate alleles revealed that the source of this diversity originated likely from nuclear-encoded plastome remnants (NUPTs). Phylogeographic and Mantel correlation analysis based on the complete chloroplast genomes exhibited clustering of individuals according to geographic distance in the first distance class, suggesting that the novel markers and in particular the cpSSRs could provide a more detailed picture of beech population structure in Central Europe.

## 1. Introduction

Chloroplasts not only play a key role in photosynthesis but also other metabolic pro-cesses of green plants [1]. The generally maternal inheritance of the chloroplast genome in Angiosperms and relatively conserved gene content and order has made chloroplast genomes a valuable resource for phylogenetic and evolutionary studies [2,3]. Plant chloroplast genomes are mostly between 120 kb and 160 kb in length and usually have a quadripartite circular structure comprising of two regions of inverted repeats A and B (IR-A/IR-B), separated by a large single-copy (LSC) region and a small single-copy (SSC) region [4]. Due to next-generation sequencing approaches, sequencing of chloroplast genomes for dozens or hundreds of individuals is now achievable through whole genome sequencing [5,6]. Chloroplast genome sequences have helped to elucidate the phylogenetic relationships and evolutionary history of many tree genera, including *Acer* [7], *Prunus* [8], *Populus* [9,10], *Quercus* [11,12] and *Pinus* [13]. With an increased availability of whole chloroplast sequences, numerous studies demonstrated the presence of variation among individuals within species, which includes SNPs, indels, inversions, translocations, copy number variations and also IR expansion, gene loss and intron retention [14]. The level of this variability is usually considered low, both in terms of composition, as well as in terms of the degree of variation within the different regions [14,15,16,17]. Interestingly, the individuals themselves have been reported to show variations of their chloroplast genomes through heteroplasmy—which can occur as a result of independent mutations or biparental inheritance of organelles in one organism [18]. While chloroplast genomes can be a good tool for phylogeographic analyses, such studies are currently limited to only few conifers [16,17].

European beech (*Fagus sylvatica* L.) is ecologically and economically one of the most important broadleaved tree species in Europe [19]. There are several molecular studies that evaluated genetic diversity and structure of European beech using chloroplast DNA [20,21,22,23,24,25,26,27]. Most of these studies showed a low level of chloroplast diversity and a rather homogeneous genetic structure in Central Europe, but none of them exploited the full potential resolution offered by complete chloroplast genomes. Thus, there is still lack of comparative analyses based on complete chloroplast genomes, which would allow to identify novel chloroplast polymorphisms and to detect genetic structure of European beech at a regional scale. Recently, Mishra and coauthors [26] evaluated three complete chloroplast genomes of beech from areas glaciated during the Weichselian glacial maximum and found a very low genetic variation with only two SNP and three indel positions. This raised the question of whether the low variation found was due to genetic empoverishment by founder effects at the leading edge of the recolonization or if chloroplast genetic diversity is generally low in European beech. To clarify this, there is a need to assess genetic diversity of complete chloroplast genome of European beech sampled from a wider range, which was the aim of the current study.

Here, we report 16 newly sequenced and assembled complete chloroplast genomes of *F. sylvatica* and perform comparative genomic analyses of the new sequences with the two recently published chloroplast genomes: Bhaga (MW531753) and Jamy (MW537046) [26]. The aim of our study was to identify potentially highly variable markers in chloroplast genome of *F. sylvatica* suitable for phylogeographic studies as a useful genetic resource for developing chloroplast-based genetic markers (SNPs and SSRs) for large-scale population studies.

## 2. Materials and Methods

### 2.1. DNA Isolation and Sequencing

Details regarding DNA isolation and sequencing of Bhaga and Jamy individuals are given in Mishra and coauthors [26]. The remaining 16 individuals representing a wide range of the species distribution were collected from Siemianice provenance trial [28]. Detailed geographic locations are presented in Table 1 and Appendix A. DNA was isolated from leaves with a GeneMATRIX Plant & Fungi DNA Purification Kit (EURx, Poland), after storing them in the dark for 48 h. Genomic library preparation and sequencing was done by an external service provider (IGA Technology Services s.r.l.) with Illumina HiSeq 2500 device in 125-bp PE mode. The obtained reads were purified from adapters and trimmed with Trimmomatic [29]. Raw reads were deposited in SRA under the accession numbers listed in Table 1.

### 2.2. Chloroplast Genome Assemblies and Annotation

Methods describing assembly and annotation of chloroplast genomes of Bhaga and Jamy individuals are presented in Mishra and coauthors [26]. All the remaining 16 chloroplast genomes where generated using the same protocol: Illumina reads were used for de novo assembly using NOVOPlasty v 4.2. [30,31] with seed sequence NCBI: AY453092.1 [32] and Bhaga chloroplast genome for guiding the program in both inverse repeat regions. After assembling, the sequences were manually checked, in case of presence of ambiguous nucleotides manual curation was done with the assistance of reads mapped to a genome with bwa-mem [33] and visualization of the results in Tablet software [34]. Coding sequences and RNA elements annotation was done with GeSeq ChloroBox [35] using chloroplast genomes of *F. crenata (*NC_041252; [36]), *F. engleriana* (KX852398; [37]) *F. japonica* (MT762294; [38]) and *F. sylvatica* (NC_041437; [39]) as references. Postprocessed annotated genomes where uploaded to GenBank, for accession numbers see Table 1.

### 2.3. Assessment of Genome Variation

REPuter was employed to identify four types of large repeating sequences (reverse, forward, complement and palindromic) with a minimum repeat size of 30 bp, hamming distance equal to 3 maximum computed repeats set to 50 [40]. Identification of chloroplast simple sequence repeats (cpSSR) was done using MISA [41]. The minimum number of repeat units was set to eight, six, five, five, three, and three, for mononucleotides, dinucleotides, trinucleotides, tetranucleotides, pentanucleotides, and hexanucleotides, respectively. For assessment of variance between the 18 studied chloroplast genomes alignments were done with MAFFT v 7.450 [42], as implemented in Unipro UGENE [43]. After this variations among the genomes where highlighted using the Bhaga chloroplast genome as reference.

### 2.4. Detection of Heteroplasmy

Potential chloroplast genome variation within individuals (heteroplasmy) was assessed with mapping of reads of each individual with bwa [33] to extracted SSC, LSC and IR regions of the genomes. The marker calling was done with Freebayes [44] with 0.02 minor allele frequency (MAF) and depth of 200x thresholds for variant detection to avoid sequencing error [45]. To verify the origin of the markers reads were mapped to chromosome 10 of the *Fagus sylvatica* nuclear assembly [46].

### 2.5. Phylogenetic Analysis

Phylogenetic analysis was done with on the basis of the dataset of the 18 *Fagus sylvatica* assemblies to which the complete chloroplast genomes of *F. crenata (*NC_041252; [36]), *F. japonica* (MT762294; [38]) and *F. engleriana* (KX852398; [37]) before alignment as described above. Phylogenetic reconstructions were done using IQ-TREE with the GTR+I+R model [47,48,49] and 1000 bootstrap replicates. The resulting phylogram was edited with Figtree 1.4.3 (http://tree.bio.ed.ac.uk/software/figtree/, accessed on 21 March 2021) with rooting to midpoint and proportional transformation of branches.

The phylogenetic distance matrix obtained from IQ-TREE was also used to test the relationship between genetic and geographic distances among individuals. The correlation was calculated with PASSaGE 2 [50] using Mantel’s test [51] for a global assessment and Mantel’s correlogram to search for significance within 10 equally paired distance classes (the largest class excluded). All tests were performed with 10,000 permutations.

## 3. Results

### 3.1. Assembly Size Variance and Genome Annotation

Read coverage of each of 16 new assemblies varied from 86x to 625x with average value of 302x and 284x median. Chloroplast genome structure was stable throughout the studied sequences, and assembly sizes varied from 158,391 bp to 158,464 bp, with highest length variance observed in the large single copy region (LSC) (87,634–87,706 bp). The small single copy regions (SSC) differed only by 4 bp (19,010–19,013 bp), while both inverted repeat regions (IR-A/IR-B) where identical in length (25,873 bp) (Table 2).

Similarly to the previously published *F. sylvatica* chloroplast assemblies [26,39] each of the 16 new genomes had an identical set of 131 annotated elements: 83 protein coding genes, 8 rRNAs and 40 tRNAs. Total share of coding elements differed across main elements of the genome, with 51% in LSC, 72.1% in SSC and 59.1% in IR regions.

### 3.2. Repeat elements and SNPs

Large repeating sequence (LRS) assessment showed that sixteen genomes had 32 LRSs >30 bp: 16 forward, 13 reverse, one complement and two palindromic. Two assemblies (Gdańsk and Glorup) had 31 LRSs as a result of the loss of one palindromic match, and one chloroplast genome (Ehingen) had 33 LRSs with an additional reverse match compared to the 16 previously mentioned assemblies. Analysis of cpSSR using MISA detected a total number of 138 markers, out of which only 4 of 97 mononucleotide cpSSRs and 8 of 35 complex cpSSRs were polymorphic. All discovered dinucleotide and pentanuclotide cpSSRs were found to be monomorphic (Table 3).

In the SSC region we found four polymorphic markers: two mononucleotide and two complex repeats. Variation of the SSR mononucleotide T at position 12,538 occurred due to the absence of a microsatellite in one of the individuals (Fantanelle). The LSC region had eight polymorphic markers with six complex and two mononucleotide cpSSR (Table 4). Marker ratio, reflecting the number of individuals associated with a particular variant, showed that in eight sites an alternative nucleotide was present, while three cpSSR loci had two, and one locus had five variants.

Alignment of the 18 chloroplast genomes revealed four SNPs and one indel located in noncoding regions. The first SNP (pos. 12,587) was associated with a mononucleotide cpSSR, the alternative variant was present in an individual from Fantanele, shortening the repetitive sequence to 7 bp. Variants for second (pos. 46,985) and fourth (pos. 112,198) SNP, as well as the indel (pos. 80,558) where present in only one individual; however, in the third SNP (pos. 71,204) 50% of individuals had the alternative variant (Table 5).

### 3.3. Within Individual Polymorphisms

Within individual polymorphism related to chloroplast genome was found in all 16 tested individuals. After filtering with and minimum depth >200x and MAF of 0,02 a total number of 9028 markers where detected in all analyzed regions.

The average depth of each base at a variant position was lower in single copy elements 349x and 360x in LSC and SSC respectively, while both IR regions had 477x depth (Table 6). However, the average alternative variant’s depth was very similar across the main genome regions from 16.1x in SSC, 18.4–18.5x in IRs and 18.7x in LSC. This suggests that these variants represent the nuclear encoded plastome sequences (NUPTs), as the these values are similar to the average coverage of 17x found on chromosome 10 of the complete nuclear genome. This chromosome was selected for comparative analysis due to lowest NUPT detection in the whole assembly.

Among the variant positions majority of them where SNPs (76.8–84.1%), the remaining share was associated to indels (8.8–10.2%), complex markers (3.1–8.2%) and MNPs (0.2–0.7%). In 2.7–4.6% of sites a mix of variants was detected e.g., a SNP and an indel called at a specific position in the same individual.

Markers detected in this study where found both in coding (48.6–67.3%) and non-coding areas (32.7–51.4%). The size of the contribution in each of these parts was related to the size of coding and non-coding regions of the main genome element (Figure 1).

### 3.4. Phylogenetic Analysis

The complete chloroplast genome sequences of 18 *F. sylvatica* individuals, as well as *F. crenata, F. japonica* and *F. engleriana*, were used to for a phylogenetic reconstruction based on maximum likelihood method and 1000 bootstrap replicates (Figure 2).

While a global Mantel’s test did not reveal significant relationship (*r* = 0.2190; *p* = 0.1861) between the phylogenetic and geographic distances for the 18 individuals, clustering of similar individuals was confirmed with Mantel’s correlogram within the first distance class (<250 km; *r* = 0.286; *p* = 0.011) (Table 7).

## 4. Discussion

Complete chloroplast genomes have helped to reveal species relationships [9,11,12], but also allow to measure divergence within populations [16,17]. Growing genomic resources for European beech provide tools to extend our knowledge on this critically important forest tree species. Our results support previous hypotheses suggesting low genetic diversity of the beech plastome [23,52,53].

The total genome length variation (158,428 ± 37 bp) and the presence of polymorphisms were associated exclusively with Single Copy (SC) dions, while the pairs of IR regions where monomorphic both in terms of the length (25,873 bp) and in terms of nucleotide sequence. Stability of IR and variability of SC regions was also present in sequences of *F.japonica* (MT762294; MT762295), the results suggest a powerful gene conversion mechanism in Fagus species. Our study revealed 138 cpSSR in *F. sylvatica* out of which 126 where monomorphic. This group included universal cpSSRs: ccmp4, ccmp7, ccmp10, commonly used to assess phylogeny and relationship in eudicot species [54,55]. Magri and coauthors [23] using these markers concluded that Central Europe beech populations generally can be considered as a homogeneous group. The 12 polymorphic microsatellite markers discovered in this study, when applied for a higher number of individuals and populations, could, however, potentially provide a more detailed phylogeographic picture. Our phylogeographic analysis supports this assumption due to significant clustering of individuals over a relatively short distance <250 km.

Additional source of variation was found in in SNPs (4) and indel (1), all located in noncoding regions, but their position is not in line with results obtained based on reduced representation genomic libraries presented by Meger and coauthors [27]. Heteroplasmy is well reported in plants with known biparental inheritance of chloroplasts, even though in some species (e.g., *Passiflora*) it can occur at the seedling and embryo but not at the mature developmental stages [56]. In beech, due to maternal inheritance of organelles [3], heteroplasmy can exclusively be caused by mutations. The evidence of multiple integrations of organelle DNA integration with the nuclear genome in beech [46] and detection of within-individual polymorphisms of cpDNA-related sequences presented in this study suggest that assessing beech diversity with chloroplast related SNPs due to a large occurrence of nuclear encoded of plastid DNA (NUPTs) can lead to uncertain results and should be taken with caution [57].

## Figures and Tables

**Figure 1 genes-12-01357-f001:**
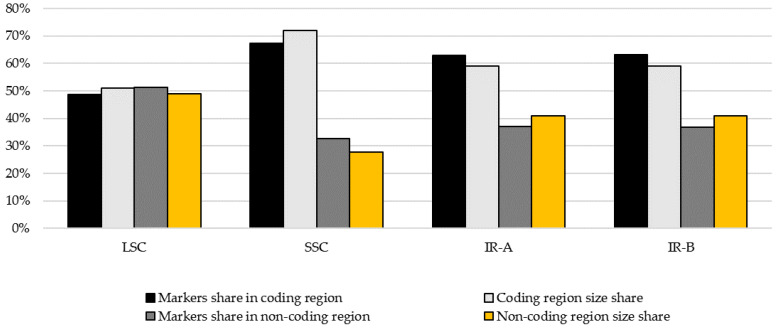
Share or markers in coding and non-coding regions in relation to the size share of coding and non-coding elements in main genome regions: LSC—large single copy region, SSC—small single copy region, IR-A/IR-B—inverted repeat regions A and B.

**Figure 2 genes-12-01357-f002:**
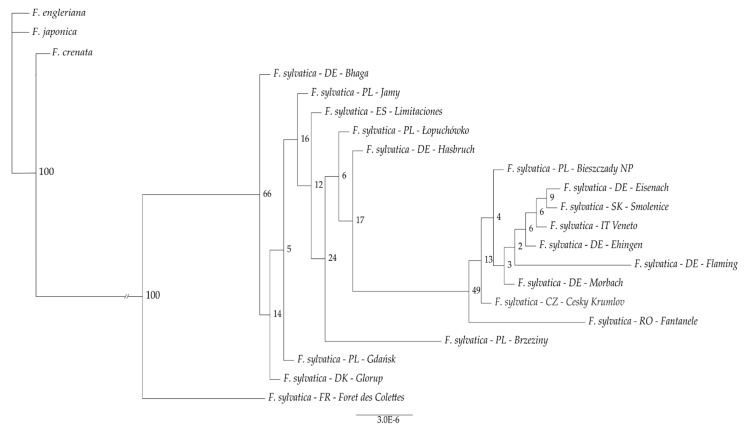
Phylogenetic relationships among 18 *F. sylvatica* individuals, as inferred using Maximum Likelihood, with *F. crenata, F. japonica*, and *F. engleriana* as outgroup. Numbers on nodes indicate percentages of bootstrap support from 1000 bootstrap replicates, the genetic distance between the *F.sylvatica* individuals and the outgroup was shorten by 0.04.

**Table 1 genes-12-01357-t001:** Origin of sampled individuals and sequencing data volume.

No.	Origin or Individual Name	Country	Longitude	Latitude	Number of Read Pairs	NCBI Accession Number	SRA Accession Number
1	Bhaga	Germany	51.169167 N	8.963056 E	[26]	MW531753	N/A
2	Jamy	Poland	53.586019 N	18.935019 E	[26]	MW537046	SAMN08948264
3	Gdańsk	Poland	54.383262 N	18.516724 E	3,777,769	MW566769	SAMN18917950
4	Foret des Colettes	France	46.183328 N	2.949992 E	4,899,373	MW566771	SAMN18917951
5	Limitaciones	Spain	42.818059 N	2.249663 W	6,210,877	MW566772	SAMN18917952
6	Glorup	Denmark	55.184748 N	10.681238 E	20,891,953	MW566770	SAMN18917953
7	Łopuchówko	Poland	52.583300 N	17.083339 E	5,114,816	MW566774	SAMN18917954
8	Hasbruch	Germany	53.120708 N	8.4302740 E	4,650,347	MW566776	SAMN18917955
9	Bieszczady NP	Poland	49.117093 N	22.579103 E	3,046,013	MW566773	SAMN18917956
10	Eisenach	Germany	50.087605 N	10.106152 E	4,461,792	MW566778	SAMN18917957
11	Morbach	Germany	50.740891 N	6.980116 E	5,833,195	MW566784	SAMN18917958
12	Ehingen	Germany	48.399106 N	9.500861 E	5,632,928	MW566775	SAMN18917959
13	Veneto	Italy	46.133489 N	12.216683 E	7,741,036	MW566783	SAMN18917960
14	Cesky Krumlov	Czechia	48.850035 N	14.250406 E	7,853,097	MW566777	SAMN18917961
15	Brzeziny	Poland	51.836489 N	19.601247 E	7,349,714	MW566779	SAMN18917962
16	Smolenice	Slovakia	48.485171 N	17.372687 E	5,072,400	MW566782	SAMN18917963
17	Fantanele	Romania	46.416750 N	26.466475 E	6,584,825	MW566780	SAMN18917964
18	Fläming	Germany	52.133389 N	12.583406 E	7,423,489	MW566781	SAMN18917965

**Table 2 genes-12-01357-t002:** Statistics for main chloroplast genome elements: LSC - large single copy region, SSC—small single copy region, IR-A/IR-B—inverted repeat regions A and B.

			Main Genome Elements			
Origin or Individual Name	Read Coverage	NCBI Accession Number	Total Size (bp)	LSC (bp)	SSC (bp)	IR-A/IR-B) (bp)
Bhaga	-	MW531753	158,458	87,702	19,010	25,873
Jamy	-	MW537046	158,462	87,705	19,011	25,873
Gdańsk	253x	MW566769	158,456	87,699	19,011	25,873
Colettes	498x	MW566771	158,391	87,634	19,011	25,873
Limitaciones	491x	MW566772	158,461	87,704	19,011	25,873
Glorup	356x	MW566770	158,461	87,704	19,011	25,873
Łopuchówko	212x	MW566774	158,461	87,704	19,011	25,873
Hasbruch	267x	MW566776	158,462	87,705	19,011	25,873
Bieszczady NP	211x	MW566773	158,426	87,669	19,011	25,873
Eisenach	105x	MW566778	158,456	87,699	19,011	25,873
Morbach	350x	MW566784	158,463	87,706	19,011	25,873
Ehingen	91x	MW566775	158,446	87,689	19,011	25,873
Veneto	625x	MW566783	158,463	87,706	19,011	25,873
Cesky Krumlov	300x	MW566777	158,462	87,705	19,011	25,873
Brzeziny	521x	MW566779	158,462	87,705	19,011	25,873
Smolenice	86x	MW566782	158,430	87,674	19,010	25,873
Fantanele	157x	MW566780	158,462	87,705	19,011	25,873
Fläming	306x	MW566781	158,464	87,705	19,013	25,873

**Table 3 genes-12-01357-t003:** General characteristics of chloroplast microsatellite markers in 18 *F. sylvatica* individuals.

	Mononucleotide	Dinucleotide	Pentanucleotide	Complex	Total
Monomorphic	93	2	4	27	126
Polymorphic	4	-	-	8	12
Total	97	2	4	35	138

**Table 4 genes-12-01357-t004:** Basic information of polymorphic chloroplast microsatellites; marker ratio—number of individuals associated with a particular marker variant; region types: LSC—Large Single Copy; SSC—Small Single Copy.

No.	Starting Position (bp) *	Type	Region	Marker Ratio	Flanking Annotation
1	4363	Complex	SSC	17/1	ndhA (exon II) ↔ ndhA (exon I)
2	8012	Complex	SSC	16/1/1	psaC ↔ ndhD
3	11,476	Mononucleotide (A)	SSC	17/1	trnL ↔ rpl32
4	12,583	Mononucleotide (T)	SSC	17/0 **	rpl32 ↔ ndhF
5	46,142	Complex	LSC	16/1/1	matK ↔ trnQ
6	46,952	Complex	LSC	11/2/2/1/1/1	matK ↔ trnQ
7	50,589	Mononucleotide (A)	LSC	17/1	trnG (exon I) ↔ trnG (exon II)
8	55,923	Complex	LSC	16/2	atpH ↔ atpI
9	70,097	Complex	LSC	16/2	rpoB ↔ trnC
10	92,043	Mononucleotide (A)	LSC	16/2	trnG (exon II) ↔ trnG (exon I)
11	105,126	Complex	LSC	12/5/1	ycf4 ↔ cemA
12	107,580	Complex	LSC	17/1	petA ↔ psbJ

* according to the Bhaga reference; ** marker absent in an individual

**Table 5 genes-12-01357-t005:** Summary of the variant sites detected in the 18 chloroplast genomes, region types: LSC—Large Single Copy; SSC—Small Single Copy.

No.	Position (bp) *	Marker Type	Region	Consensus	Alternative	Area	Marker Ratio	Flanking Annotation
1	12,587	SNP	SSC	T	C	noncoding	17/1	rpl32 ↔ ndhF
2	46,985	SNP	LSC	G	A	noncoding	17/1	tRNA-K ↔tRNA-Q
3	71,204	SNP	LSC	G	T	noncoding	9/9	tRNA-C ↔ petN
4	80,558	Indel	LSC	T	-	noncoding	17/1	psbZ ↔ tRNA-G
5	112,198	SNP	LSC	A	C	noncoding	17/1	psaJ ↔ rpl3

* the position (bp) is referred to the Bhaga genome

**Table 6 genes-12-01357-t006:** Summary statistics of within individual polymorphisms detected in regions of the 16 chloroplast genome assemblies. LSC - large single copy region, SSC—small single copy region, IR-A/IR-B—inverted repeat regions A and B.

	LSC	SSC	IR-A	IR-B
Avg. variant depth	349x	360x	477x	477x
Avg. alternative var. depth	18.7x	16.1x	18.4x	18.5x
Number of uniqe positions	5348	1161	1257	1262
SNP	76.8%	80.9%	83.7%	84.1%
Indel	10.2%	8.8%	9.6%	9.4%
Complex	8.2%	6.2%	3.1%	3.1%
MNP	0.2%	0.3%	0.8%	0.7%
Mix	4.6%	3.9%	2.7%	2.7%
Coding	48.6%	67.3%	62.9%	63.1%
Non-coding	51.4%	32.7%	37.1%	36.9%

**Table 7 genes-12-01357-t007:** Summary of Mantel’s test statistics calculated within consecutive distance classes.

Class	Boundry max (km)	Number of Pairs	Mantel r	*p*
1	250	11	0.286	0.011
2	500	31	0.106	0.361
3	750	46	0.121	0.144
4	1000	27	−0.016	0.760
5	1250	15	−0.004	0.900
6	1500	11	−0.023	0.374

## Data Availability

The chloroplast genome sequences have been deposited in GenBank under the accession numbers: MW566769, MW566771, MW566772, MW566770, MW566774, MW566776, MW566773, MW566778, MW566784, MW566775, MW566783, MW566777, MW566779, MW566782, MW566780, MW566781.

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
