# Peer review of "Complete Chloroplast Genomes of *Fagus sylvatica* L. Reveal Sequence Conservation in the Inverted Repeat and the Presence of Allelic Variation in NUPTs"

_genes, 2021, doi:10.3390/genes12091357_

Round 1

Reviewer 1 Report

The article entitiled "Complete chloroplast genomes of Fagus sylvatica L. reveal sequence conservation in the inverted repeat and the presence of
allelic variation in NUPTs. " is nice research article. It is well designed and nicely written. The manucript is acceptable in the present form with following minor changes.

Comment1# Try to avoid the use of abbreviation, which are used in the text only one time for example :In line 47 NGS, CNVs in 54 , IRA in line 105 etc.

Comment2# In  line 109: Write the Fagus crenata, Fagus engleriana as F. crenata, F engleriana as the full name is already mentioned in introduction.

Comment3# Redraw fig 2 by increasing font size and here also write "F." only instead of "Fagus" in all names.

Comment4# In line101 , rewrite the sentence " Details on the chloroplast genome .......Mishra et al. [26]. 

Comment5# In line 114 put the citation in the last of the sentence.

Author Response

Reviewer: Comment1# Try to avoid the use of abbreviation, which are used in the text only one time for example :In line 47 NGS, CNVs in 54 , IRA in line 105 etc.

Authors: Thank you for pointing this out, all the abbreviations applied only once, were removed.

Reviewer: Comment2# In  line 109: Write the Fagus crenata, Fagus engleriana as F. crenata, F engleriana as the full name is already mentioned in introduction.

Authors: Corrected.

Reviewer: Comment3# Redraw fig 2 by increasing font size and here also write "F." only instead of "Fagus" in all names.

Authors: The names were corrected, the font size was increased by 2 points.

Reviewer: Comment4# In line101 , rewrite the sentence " Details on the chloroplast genome .......Mishra et al. [26]. 

Authors: The sentence was rewritten to: Details regarding DNA isolation and sequencing of Bhaga and Jamy individuals are given in Mishra and coauthors [26].

Reviewer: Comment5# In line 114 put the citation in the last of the sentence.

Authors: Corrected.

Reviewer 2 Report

Assessing the genetic diversity of the complete chloroplast genome of European beech is interesting to the readers and novelty for the science.
The results of the study are useful for future phylogeographic and large-scale population studies.  

Line 21: Double comma;

Line 30: Double point;

Line 51: Pinus taeda L. is not a genus;

When quoting an author team, it is more correct to write Mishra and coauthors instead of Mishra et al. (E.g. Line 70).

Line 73: thelow is written merged;

The decimal point is also written "." in other places, "," (for example, line: 214).

Author Response

Reviewer: Line 21: Double comma;

Authors: Corrected.

Reviewer: Line 30: Double point;

Authors: Corrected.

Reviewer: Line 51: Pinus taeda L. is not a genus;

Authors: Corrected to Pinus.

Reviewer: When quoting an author team, it is more correct to write Mishra and coauthors instead of Mishra et al.(E.g. Line 70).

Authors: Thank you for pointing this out, it was corrected in the whole text.

Reviewer: Line 73: thelow is written merged;

Authors: Corrected.

Reviewer: The decimal point is also written "." in other places, "," (for example, line: 214).

Authors: Corrected.

Reviewer 3 Report

This study is about chloroplast genome information of European beech(Fagus sylvatica). As the authors point out, the accumulation of chloroplast genome information will provide useful information for phylogeographic studies of European beech. Here are some comments

In Line65-70, the authors state that "Most of these studies...beech at a regional scale.” However, the bootstrap probability of Fig. 2 is not necessarily high, and it may be difficult to reveal the genetic structure at the regional scale even if the complete chloroplast genome is used.

There are no results or descriptions that address these questions described in Lines 72-75.

The "highly variable regions in the chloroplast genome" shown in Lines 81-82 are described, but there is no evidence that the "highly variable regions" have been clarified.

The description in Line 255-260 is only a hypothesis. Furthermore, the bootstrap probability of the phylogenetic tree shown in Fig. 2 is low, and there is not enough information to clarify the divergence.

I have read this paper many times and have not found any results that answer the authors' phylogeographic research assumptions. The authors should review the paper again and reconsider the significance of this study. To me, the results on heteroplasmy are rather interesting, and I think it would be better to reconsider the paper on the subject of heteroplasmy. The impact of Heteroplasmy will alert readers to new caveats to SNP analysis using chloroplast genomes.

The sentence in line 1 should be deleted. The sentence in line 1 should be deleted because physiological studies are not described in this paper.

Author Response

Reviewer: In Line65-70, the authors state that "Most of these studies...beech at a regional scale.” However, the bootstrap probability of Fig. 2 is not necessarily high, and it may be difficult to reveal the genetic structure at the regional scale even if the complete chloroplast genome is used.

 Authors: We thank for this comment, we are aware that the boostrap levels are low, but the results indicating genetic clustering are supported by results in Table 7.

Reviewer: There are no results or descriptions that address these questions described in Lines 72-75.

Authors: The results clearly show that wide-range diversity of beech chloroplast genome is low, due to rather small scale of this study we did not want to unequivocally decide about the nature of this fact.

Reviewer: The "highly variable regions in the chloroplast genome" shown in Lines 81-82 are described, but there is no evidence that the "highly variable regions" have been clarified.

Authors: Yes, you are right, changed from “regions” to “markers”.

Reviewer: The description in Line 255-260 is only a hypothesis. Furthermore, the bootstrap probability of the phylogenetic tree shown in Fig. 2 is low, and there is not enough information to clarify the divergence.

 Authors: During the analysis we have tested higher bootstrap replicated levels >10.000 and the results in general are the same. Although we indicated some level of local clustering of beech individuals our main result is that beech diversity is low.

Reviewer: I have read this paper many times and have not found any results that answer the authors' phylogeographic research assumptions. The authors should review the paper again and reconsider the significance of this study. To me, the results on heteroplasmy are rather interesting, and I think it would be better to reconsider the paper on the subject of heteroplasmy. The impact of heteroplasmy will alert readers to new caveats to SNP analysis using chloroplast genomes.

 Authors: We thank for this comment, our study has limitations related to the number of individuals and their unequal spatial distribution. We believe that the results shed additional light on the phylogeography of the species. While at the same time in the newly prepared paper we would like to significantly extend this problem on a much larger scale, taking into account data related to both the nuclear and chloroplast genomes. When preparing this publication, we also realized the importance of NUPTs, and therefore we are preparing a separate publication that will shed light on this problem, taking into account many species, not only beech.

Reviewer: The sentence in line 1 should be deleted. The sentence in line 1 should be deleted because physiological studies are not described in this paper.

Authors: Corrected, we have deleted this line.